# Gaps in the HIV diagnosis and care cascade for migrants in Australia, 2013–2017: A cross-sectional study

Tafireyi Marukutira[1,2]*, Richard T. Gray[3], Caitlin Douglass[1,4], Carol El-Hayek[1], Clarissa Moreira[1], Jason Asselin[1], Basil Donovan[3,5], Tobias Vickers[3], Tim Spelman[1], Suzanne Crowe[1,6], Rebecca Guy[3], Mark Stoove[1,2], Margaret Hellard[1,2]

1 Public Health Discipline, Burnet Institute, Melbourne, Australia, 2 School of Public Health and Preventive Medicine, Monash University, Melbourne, Australia, 3 The Kirby Institute, UNSW Sydney, Sydney, Australia, 4 School of Population and Global Health, University of Melbourne, Melbourne, Australia, 5 Sydney Sexual Health Centre, Sydney, Australia, 6 Department of Infectious Diseases, Monash University, Melbourne, Australia

* Tafireyi.marukutira@burnet.edu.au

**Data Availability Statement:** All relevant data are within the manuscript and its Supporting Information files.

## Abstract

### Background

Globally, few studies compare progress toward the Joint United Nations Program on HIV/AIDS (UNAIDS) Fast-Track targets among migrant populations. Fast-Track targets are aligned to the HIV diagnosis and care cascade and entail achieving 90-90-90 (90% of people living with HIV [PLHIV] diagnosed, 90% of those diagnosed on treatment, and 90% of those on treatment with viral suppression [VS]) by 2020 and 95-95-95 by 2030. We compared cascades between migrant and nonmigrant populations in Australia.

### Methods and findings

We conducted a serial cross-sectional survey for HIV diagnosis and care cascades using modelling estimates for proportions diagnosed combined with a clinical database for proportions on treatment and VS between 2013–2017. We estimated the number of PLHIV and number diagnosed using New South Wales (NSW) and Victorian (VIC) data from the Australian National HIV Registry. Cascades were stratified by migration status, sex, HIV exposure, and eligibility for subsidised healthcare in Australia (reciprocal healthcare agreement [RHCA]). We found that in 2017, 17,760 PLHIV were estimated in NSW and VIC, and 90% of them were males. In total, 90% of estimated PLHIV were diagnosed. Of the 9,391 who were diagnosed and retained in care, most (85%; *n* = 8,015) were males. We excluded 38% of PLHIV with missing data for country of birth, and 41% (*n* = 2,408) of eligible retained PLHIV were migrants. Most migrants were from Southeast Asia (SEA; 28%), northern Europe (12%), and eastern Asia (11%). Most of the migrants and nonmigrants were males (72% and 83%, respectively). We found that among those retained in care, 90% were on antiretroviral therapy (ART), and 95% of those on ART had VS (i.e., 90-90-95). Migrants had larger gaps in their HIV diagnosis and care cascade (85-85-93) compared with

**Funding:** The authors received no specific funding for this work. TM and CD are supported by an Australian Government Research Training Program (RTP) Scholarship for their PhD studies. MH receives a Fellowship from the National Health and Medical Research Council of Australia. The ACCESS project is funded by the Australian Department of Health. The Kirby Institute receives funding from the Australian Government Department of Health and is affiliated with the Faculty of Medicine, UNSW Sydney, Australia.

**Competing interests:** The authors have declared that no competing interests exist.

**Abbreviations:** ABS, Australian Bureau of Statistics; ACCESS, Australian Collaboration for Coordinated Enhanced Sentinel Surveillance of Sexually Transmissible Infections and Blood Borne Viruses; ART, antiretroviral therapy; ECDC, European Centre for Disease Prevention and Control; IRR, incidence rate ratio; MSM, men who have sex with men; NSW, New South Wales; PLDHIV, people living with diagnosed HIV; PLHIV, people living with HIV; RHCA, reciprocal healthcare agreement; SEA, Southeast Asia; SSA, sub-Saharan Africa; UNAIDS, Joint United Nations Program on HIV/AIDS; VIC, Victorian; VL, viral load test; VS, viral suppression.

nonmigrants (94-90-96). Similarly, there were larger gaps among migrants reporting male-to-male HIV exposure (84-83-93) compared with nonmigrants reporting male-to-male HIV exposure (96-92-96). Large gaps were also found among migrants from SEA (72-87-93) and sub-Saharan Africa (SSA; 89-93-91). Migrants from countries ineligible for RHCA had lower cascade estimates (83-85-92) than RHCA-eligible migrants (96-86-95). Trends in the HIV diagnosis and care cascades improved over time (2013 and 2017). However, there was no significant increase in ART coverage among migrant females (incidence rate ratio [IRR]: 1.03; 95% CI 0.99–1.08; $p = 0.154$), nonmigrant females (IRR: 1.01; 95% CI 0.95–1.07; $p = 0.71$), and migrants from SEA (IRR: 1.03; 95% CI 0.99–1.07; $p = 0.06$) and SSA (IRR: 1.03; 95% CI 0.99–1.08; $p = 0.11$). Additionally, there was no significant increase in VS among migrants reporting male-to-male HIV exposure (IRR: 1.02; 95% CI 0.99–1.04; $p = 0.08$). The major limitation of our study was a high proportion of individuals missing data for country of birth, thereby limiting migrant status categorisation. Additionally, we used a cross-sectional instead of a longitudinal study design to develop the cascades and used the number retained as opposed to using all individuals diagnosed to calculate the proportions on ART.

## Conclusions

HIV diagnosis and care cascades improved overall between 2013 and 2017 in NSW and VIC. Cascades for migrants had larger gaps compared with nonmigrants, particularly among key migrant populations. Tracking subpopulation cascades enables gaps to be identified and addressed early to facilitate achievement of Fast-Track targets.

## Author summary

### Why was this study done?

- Reaching the end of the AIDS epidemic by achieving the Joint United Nations Program on HIV/AIDS (UNAIDS) Fast-Track targets requires tracking progress across all population groups.

- Fast-Track targets entails reaching 90-90-90 (90% of people living with HIV diagnosed, 90% of those diagnosed on treatment, and 90% of those on treatment with viral suppression) by 2020 and 95-95-95 by 2030.

- Migrants often face barriers to healthcare access because of their migration status, sociocultural influence, financial constraints, and stigma.

- There are limited data on HIV diagnosis, care, and treatment among migrants, which is required in order to streamline the HIV response.

### What did the researchers do and find?

- We developed HIV diagnosis and care cascades for migrants and nonmigrants in Australia between 2013 and 2017.

- The overall cascade was high in New South Wales and Victoria (i.e., 90-90-95).

- Migrants had inferior cascades (85-85-93) compared with nonmigrants (94-90-96), including migrants reporting male-to-male HIV exposure (84-83-93) compared with nonmigrants (96-92-96).

- Between 2013 and 2017, subpopulation cascades improved, except for females, irrespective of migration status and migrants from Southeast Asia.

### What do these findings mean?

- In our study, although the overall HIV diagnosis and care cascades improved over time in Australia, migrants lagged behind.

- Failure to develop subpopulation cascades may lead to gaps going unnoticed.

- To address gaps, there is need for streamlined strategies, especially among migrant populations, to ensure timely HIV testing and linkage to care.

## Introduction

Reaching the end of the AIDS epidemic by achieving the Joint United Nations Program on HIV/AIDS (UNAIDS) Fast-Track targets set for 2020 and 2030 [1,2] requires tracking progress across all populations at risk and living with HIV. Fast-Track targets entail 90% of people living with HIV (PLHIV) are diagnosed, 90% of those diagnosed receive sustained antiretroviral therapy (ART), and 90% of those on ART achieve viral suppression (VS) (i.e., 90-90-90) by 2020 and scaling up to 95-95-95 by 2030. Reaching the Fast-Track targets by 2030 is projected to lead to at least 90% reduction of HIV incidence compared with 2010 [2].

Although progress is being made globally, significant gaps still exist as we work toward the UNAIDS target; importantly, new HIV infections are not decreasing fast enough [3]. The incidence reduction goal is more likely to be achieved if HIV prevention strategies are strengthened [4] and if the potential pools of high HIV viral loads are eliminated. In 2017, 75% of PLHIV globally were diagnosed, 79% of those diagnosed were on ART, and 81% of those were on ART with VS (i.e., 59% on ART and 48% virally suppressed of all PLHIV). However, there were 1.8 million new infections, which was higher than anticipated [5]. It is thought the higher-than-expected incidence may be due to gaps in key populations receiving care, including adolescent girls, young women, and migrants [5].

Migrants comprise a considerable proportion of PLHIV globally, but there are limited data on diagnosis and treatment rates in this important population. Although the UNAIDS Global AIDS update report provides HIV diagnosis and care cascade data at global, regional, and country levels, the onus is upon countries to track subpopulation cascades. In Australia, 29% of the population was born overseas, and 33% of the 27,545 estimated to be living with HIV in 2017 were born overseas—mainly Southeast Asia (SEA), sub-Saharan Africa (SSA), and the Americas [6,7]. In 2017, 89% of PLHIV in Australia were aware of their HIV status, 87% were on ART, and of those, 95% were virally suppressed [6]. Although the number of new infections decreased in Australia, particularly among men who have sex with men (MSM), new HIV infections among migrants increased [6]. Migrants were also more likely to present with a late HIV diagnosis than citizens [6, 8,9]. These differences raise concerns that migrants are being

'left behind', highlighting the need to focus on this group if Australia is to achieve the Fast-Track targets.

Migrants face barriers to healthcare access because of their migration status, sociocultural influence, financial constraints, and stigma [5,10]. Limited access to healthcare for migrants has been highlighted in many European countries, including Spain, Portugal, and Ireland, with the need to consider equity and equality [11,12]. In Asia, migrant workers had limited access to HIV testing, especially when undocumented [13]. In Australia, citizens and permanent residents have subsidised general healthcare access, including HIV care services, through the national health insurance scheme (Medicare). Additionally, temporary residents from countries with a reciprocal healthcare agreement (RHCA) with Australia (i.e., Belgium, Finland, Italy, Malta, Netherlands, New Zealand, Norway, Ireland, Slovenia, Sweden, and United Kingdom) are eligible for subsidised access through Medicare. However, migrants outside of these categories are not eligible for Medicare, although some may access HIV care and treatment through compassionate access programmes or private health insurance [6,8]. When reporting HIV diagnosis and care cascades, it is important to provide subpopulation cascades to track progress toward Fast-Track targets and ensure no one is left behind.

In this study, we developed the HIV diagnosis and care cascades for migrants (stratified according to migrant populations) and compared them to the cascade of nonmigrants (2013–2017) to explore gaps and opportunities to inform the national and state HIV response strategies.

## Methods

We developed HIV diagnosis and care cascades using modelling estimates for proportions diagnosed combined with a clinical database for proportions on treatment and virally suppressed between 2013–2017 following a prespecified analysis plan (S1 Text) and the STROBE checklist for observational studies (S1 STROBE Checklist).

### Study population and data sources

To estimate the number of people living with diagnosed HIV (PLDHIV) we primarily used New South Wales (NSW) and Victorian (VIC) data from the Australian National HIV Registry. Additional data from the Australian Bureau of Statistics (ABS) and follow-up studies of people recently diagnosed with HIV were used to inform population movement estimates [14]. The number of undiagnosed PLHIV was estimated using back-projection modelling from CD4+ count at diagnosis [15].

In Australia, all new HIV diagnoses must be reported by laboratories and/or doctors to state and territory health departments, who submit these data to the National HIV Registry [6,16]. State and territory authorities use this information to analyse and report surveillance data to the National HIV Registry, which is maintained by the Kirby Institute at the University of New South Wales (UNSW Sydney), on behalf of the Australian Government Department of Health [17]. HIV diagnoses among immigrants to Australia are added to the national HIV registry if they have lived in Australia for at least 3 months and intend to reside in the notifying state/territory, including among people previously diagnosed overseas [6]. HIV notifications in the registry are accompanied by a range of information with varying levels of completeness.

Data from the Australian Collaboration for Coordinated Enhanced Sentinel Surveillance of Sexually Transmissible Infections and Blood Borne Viruses (ACCESS) database were used to determine the number retained in care and calculate proportions on treatment and VS [18]. ACCESS is a national sexual health surveillance network that collects and collates routine deidentified data from over 120 sexual health clinics, general practice clinics, community health

services, hospitals, and pathology laboratories across Australia, and records are anonymously linked between sites [18]. The ACCESS network collects demographic, testing, diagnosis, and treatment data to monitor the sexual health of high-risk populations. Compared with other states, NSW and VIC have the highest coverage in the ACCESS database, and they comprise the highest proportion of PLHIV and migrants in Australia [6,7].

ACCESS data have been used previously to report on HIV treatment outcomes mostly in NSW and VIC [14,19]. The ACCESS database has a high coverage of treatment and viral load data for PLHIV in NSW and VIC. A total of 7,998 and 9,727 PLHIV were captured in the ACCESS network in 2016 and 2017, respectively, across seven general practice clinics, two hospitals, and 33 sexual health clinics and linked. ACCESS captures 53%–57% of the estimated PLHIV (diagnosed and nondiagnosed) and 59%–63% of the estimated diagnosed PLHIV in NSW and VIC [14,20].

## Data collection

Data collected in ACCESS include date of visit, country of birth, area of residence, sex, self-reported HIV transmission category, ART status, and HIV viral load results. Country of birth is self-reported to the diagnosing clinician; we used this variable to classify individuals as migrants (born outside Australia) or nonmigrants (born in Australia). Based on country of birth, individuals were also grouped into region of birth based on the ABS classification [7]. Country of birth was also used to determine whether the individual was from a country eligible for RHCA with Australia [21]. Individuals missing country of birth and sex were compared separately. We excluded trans and gender-diverse persons because of their small numbers (6 in 2013 and 18 in 2017).

**Cascade variables.** We produced estimates for the HIV diagnosis and care cascade for the overall population and by migration status, sex, male-to-male HIV exposure, migrants from SEA and SSA, and migrants from countries with RHCA with Australia.

The first two steps of the cascades (the number of PLHIV and the number of diagnosed PLHIV) were estimated using previously developed methods applied to data from Australia's National HIV Registry, with numbers rounded to the nearest 10. Full details of the registry and calculations are available from the HIV, viral hepatitis, and sexually transmissible infections in Australia Annual Surveillance Report, 2018 [6]; previous publications [14,22]; and an online repository containing all the code used in the calculations [23].

To estimate the number of diagnosed PLHIV for each stratification, complete notification data for several key variables were required. To address the missing data in the HIV Registry and to obtain the complete notification data required for our analysis, we used predictive mean matching univariate imputation using known notification data to the end of 2017 from the six notification variables: year of diagnosis (100% complete), age at diagnosis (99.4% complete), sex and gender (99.5% complete), country and region of birth (69.6% complete), and exposure category/likely route of HIV transmission (86.6% complete) for each jurisdiction (100% complete). From this imputation process, we produced 10 sets of notifications with complete information for each of the six variables to account for statistical uncertainty. This imputation was conducted using the MICE software package in the R statistical software program [24,25]. Using the 10 imputed notifications sets, we estimated the number of PLHIV and the number diagnosed in NSW and VIC to 31 December 2017 for each region of birth, with an estimated upper and lower bound for each, as described in the following two subsections.

## PLDHIV (first 90)

From each imputed set of notifications, we determined the annual number of PLDHIV for each subpopulation in NSW and VIC separately by summing all notifications meeting the

appropriate criteria and then subtracting duplicates, estimated deaths, overseas emigrants, and net interstate departures. The range for PLDHIV for each set was estimated by multiplying the appropriate lower and upper bounds for the number of deaths (estimated using the 95% CI of the death rate), emigrants, and interstate arrivals and departures. Our overall estimate for PLDHIV was then obtained by summing the mean NSW and VIC estimates for each population from the 10 imputed notification sets with a range between the minimum and maximum values from the sets.

**Number of undiagnosed PLHIV.** To estimate the proportion of PLHIV who were undiagnosed for each population, we applied the European Centre for Disease Prevention and Control (ECDC) HIV Modelling Tool and added this to the estimated number of PLDHIV [15]. The ECDC tool is a multistate back-calculation model using HIV notification data and estimates for the rate of CD4+ T-cell decline to fit diagnosis rates over time, which produces an estimate and range of the undiagnosed population and the percentage undiagnosed. The range in PLHIV was obtained by applying the lower estimate for proportion undiagnosed to the lower limit for PLDHIV and, conversely, for the upper bound.

$$\text{First } 90 = \frac{\text{Number of estimated PLDHIV}}{\text{Number of estimated PLHIV}}$$

## Number of PLDHIV retained in care

We used the ACCESS database to identify the number retained in care, defined as PLDHIV with a consult/HIV viral load test (VL)/on ART in that year. Using the ACCESS database for those retained in care, we calculated the proportions on ART and virally suppressed to complete the HIV diagnosis and care cascade.

## Proportion of PLDHIV retained in care on ART (second 90)

We used ART status as captured in the ACCESS database for PLDHIV receiving a three-drug regimen for treatment in that year. We queried the database for all HIV-positive individuals with ART documentation per year. We used this proportion to estimate the second 90 instead of the traditional proportion of diagnosed who are on ART.

$$\text{Second } 90 = \frac{\text{Number of PLDHIV retained in care on ART in ACCESS database}}{\text{Number of PLDHIV retained in care in ACCESS database}}$$

## Proportion of PLDHIV retained in care on ART with VS (third 90)

To calculate the number of PLDHIV retained in care on ART with VS, we used the last VL in the year for patients on ART recorded in the ACCESS database irrespective of ART duration. VS was defined as VL < 200 copies per millilitre.

$$\text{Third } 90 = \frac{\text{Number of PLDHIV retained in care on ART with VS in ACCESS database}}{\text{Number of PLDHIV retained in care on ART in ACCESS database with a VL result}}$$

## Statistical methods

To build the HIV testing and treatment cascade for each year, proportions were calculated for the first 90, second 90, and third 90 based on the UNAIDS 90-90-90 targets (as described previously). The cascades were developed using a cross-sectional approach applied annually.

We developed the overall cascade and subpopulation cascades by sex, migration status, male-to-male HIV exposure, migrants born in SEA, migrants born in SSA, and RHCA status.

We estimated the overall ART coverage (first 90 * second 90) and VS (first 90 * second 90 * third 90) among all estimated PLHIV and compared it with targets of 81% and 73% by 2020, respectively, by calculating the appropriate proportion and the 95% CI for the proportions [2].

Individuals missing country of birth (could not be classified as migrants or nonmigrants) or sex in the ACCESS database were tracked separately for ART coverage and VS.

Poisson regression was used to describe annual trends of cascade proportions over time (2013–2017) using incidence rate ratio (IRR), 95% CI, and a *p*-value. Data modelled in the Poisson regression were checked for overdispersion. Data analysis was conducted using Stata IC 14 (Stata Corp, College Station, TX), with the significance level set at $p < 0.05$. We present the cascades for 2017 in detail and used all cascades (2013–2017) to explore trends.

The study was approved by The Alfred Ethics Committee (Project No. 248/17) and Monash University Human Ethics Committee (Project No. 11221).

## Results

### HIV diagnosis and care cascade in 2017

In 2017, there were an estimated 17,760 PLHIV in NSW and VIC; 90% were males, and 90% (95% CI 90%–91%) had been diagnosed (PLDHIV). Of the 9,391 retained in care in the ACCESS database in 2017, most (85%, *n* = 8,015) were males. Migration status could not be determined for 38% of PLHIV because of missing country of birth. Of those with a country of birth entry (*n* = 5,817), 41% (*n* = 2,408) were migrants. Most migrants were from SEA (28%), northern Europe (12%), and eastern Asia (11%). Most of the migrants and nonmigrants were males (72% and 83%, respectively). Of the retained migrants, 90% (95% CI 90%–91%) were on ART, and 95% (95% CI 95%–96%) of those on ART had VS. Overall, we estimated that 81% and 77% of all estimated PLHIV were on ART and virally suppressed, respectively (Table 1). Of the 38% of PLHIV missing country of birth in the ACCESS database, 94% (95% CI 94%–95%) were on ART, and 96% (95% CI 95%–97%) on ART had VS. Of those missing sex (9%), 96% (95% CI 95%–98%) were on ART, and 96% (95% CI 95%–97%) on ART had VS. The rest of the data used to develop the HIV care cascades and the estimated numerators and denominators for the cascade calculations are included in the Supporting information (S1 and S2 Tables).

Migrant subpopulations had larger gaps in the HIV diagnosis and care cascade estimates compared with nonmigrants, as shown in Table 1. Among migrants estimated to be living with HIV (37%; *n* = 6,570), 85% (95% CI 85%–86%) were diagnosed, and of those retained in care, 85% (95% CI 84%–87%) were on ART, and 93% (95% CI 92%–94%) of those on ART had VS. Among nonmigrants estimated to be living with HIV, 94% (95% CI 93%–94%) were diagnosed, and of those retained in care, 90% (95% CI 89%–91%) were on ART, and 96% (95% CI 95%–97%) had VS.

The estimated overall ART coverage was the highest among nonmigrants (88%), nonmigrant males (84%), migrants born in SSA (83%), and those eligible for RHCA (83%) and lowest among nonmigrant females (63%). The estimated overall VS was >73% in migrants born in SSA (75%), migrants with RHCA (74%), nonmigrant males (76%), and nonmigrants reporting male-to-male HIV exposure (80%) (Table 1).

Migrants reporting male-to-male HIV exposure had lower cascade estimates (84-87-93) compared with nonmigrants reporting similar exposure (96-92-96). Migrants born in SEA also had lower cascade estimates (72-87-93) compared with those born in SSA (89-93-91). Similarly, migrants from countries not eligible for RHCA had lower cascade estimates (83-85-92) compared with those eligible (96-86-95). Females had lower cascade estimates irrespective of whether they were migrants (87-79-90) or nonmigrants (92-70-88) (Table 1).

**Table 1. HIV diagnosis and care cascades in New South Wales and Victoria (2017).**

| Variable | First 90, % (95% CI) | Second 90, % (95% CI) | Third 90, % (95% CI) | Estimated on ART of all PLHIV* | Estimated viral suppression of all PLHIV** |
|---|---|---|---|---|---|
| **All PLHIV** | 90 (90–91)[¥] | 90 (90–91) | 95 (95–96)[¥] | 81[£] | 77[£] |
| Male | 90 (90–91)[¥] | 91 (90–91) | 95 (95–96)[¥] | 82[£] | 79[£] |
| Female | 89 (87–90) | 78 (74–81) | 91 (87–94) | 69 | 63 |
| **Migrants** | 85 (85–86) | 85 (84–87) | 93 (92–94)[¥] | 73 | 67 |
| Male | 85 (84–86) | 84 (83–86) | 92 (91–94)[¥] | 71 | 66 |
| Female | 87 (85–89) | 79 (74–84) | 90 (84–94) | 69 | 62 |
| **Nonmigrant** | 94 (93–94)[¥] | 90 (89–91) | 96 (94–96)[¥] | 85[£] | 81[£] |
| Male | 93 (93–94)[¥] | 90 (88–91) | 96 (95–97)[¥] | 84[£] | 80[£] |
| Female | 92 (90–94)[¥] | 70 (63–77) | 88 (80–94) | 64 | 57 |
| **Male-to-male HIV exposure** | 91 (90–91)[¥] | 90 (89–91) | 96 (95–96)[¥] | 82[£] | 79[£] |
| Migrant | 84 (83–85) | 87 (85–88) | 93 (91–94)[¥] | 73 | 68 |
| Nonmigrant | 96 (95–96)[¥] | 92 (90–93) | 96 (95–97)[¥] | 88[£] | 85[£] |
| **Migrants born in SEA** | 72 (70–74) | 87 (84–90) | 93 (90–95)[¥] | 63 | 58 |
| **Migrants born in SSA** | 89 (87–91) | 93 (89–95) | 91 (87–95) | 83[£] | 75[£] |
| **Migrants eligible for RHCA** | 96 (95–97) | 86 (82–88) | 95 (92–97)[¥] | 83[£] | 78[£] |
| **Migrants not eligible for RHCA** | 83 (82–84) | 85 (84–87) | 92 (91–94)[¥] | 71 | 65 |
| **Missing country of birth** | n/a | 94 (94–95) | 96 (95–97) | n/a | n/a |
| **Missing sex** | n/a | 96 (95–98) | 96 (95–97) | n/a | n/a |

*The target for estimated proportion on ART of all PLHIV in 2020 is 81%, and estimates were calculated per subpopulation living with HIV.

**The target for estimated proportion with viral suppression of all PLHIV is 73% in 2020, and estimates were calculated per subpopulation living with HIV.

[¥]90-90-90 target met.

[£]Overall target met.

Abbreviations: ART, antiretroviral therapy; PLHIV, people living with HIV; RHCA, reciprocal healthcare agreement; SEA, Southeast Asia; SSA, sub-Saharan Africa

## HIV diagnosis and care cascade trends (2013–2017)

Between 2013–2017, the trend in the overall annual proportion diagnosed (first 90) in NSW and VIC was stable (IRR: 1.00; 95% CI 0.99–1.01; $p$ = 0.281), including among migrants (IRR: 1.01; 95% CI 0.99–1.02; $p$ = 0.142), nonmigrants (IRR: 1.00; 95% CI 0.99–1.01; $p$ = 0.71), and other subpopulations (Table 2).

Fig 1 highlights the cascade trends of migrants compared with nonmigrants, and Fig 2 compares male-to-male HIV exposures. The trend in the overall annual proportion on ART (second 90) increased (IRR: 1.06; 95% CI 1.05–1.06; $p$ < 0.001), including in migrants (IRR: 1.04; 95% CI 1.03–1.06; $p$ < 0.001) and nonmigrants (IRR: 1.04; 95% CI 1.03–1.06; $p$ < 0.001) (Fig 1). However, there was no increasing trend among migrant females (IRR: 1.03; 95% CI 0.99–1.08; $p$ = 0.154), nonmigrant females (IRR: 1.01; 95% CI 0.95–1.07; $p$ = 0.707), migrants from SEA (IRR: 1.03; 95% CI 0.99–1.07; $p$ = 0.055), and migrants from SSA (IRR: 1.03; 95% CI 0.99–1.08; $p$ = 0.108) (Table 2).

VS proportions were generally high over the years and were ≥90% in 2017 in all subpopulations except nonmigrant females. There was an increasing trend in VS among all PLHIV (IRR: 1.02; 95% CI 1.01–1.03; $p$ < 0.001). However, there was no increasing VS trend in migrants reporting male-to-male HIV exposure (IRR: 1.02; 95% CI 0.99–1,04; $p$ = 0.076) (Fig 2), female migrants (IRR: 1.02; 95% CI 0.96–1.08; $p$ = 0.584), migrants born from SEA (IRR: 1.03; 95% CI 0.99–1.06; $p$ = 0.178), and migrants born from SSA (IRR: 1.01; 95% CI 0.96–1.06; $p$ = 0.794) (Table 2).

**Table 2. HIV diagnosis and care cascade trends for New South Wales and Victoria (2013–2017).**

| Variable | 2013 (%) | 2014 (%) | 2015 (%) | 2016 (%) | 2017 (%) | IRR (95% CI) | *p*-Value |
|---|---|---|---|---|---|---|---|
| **Diagnosed (first 90)** | 89 | 90 | 90 | 90 | 90 | 1.00 (0.99–1.01) | 0.281 |
| Male | 89 | 90 | 90 | 90 | 90 | 1.00 (0.99–1.01) | 0.444 |
| Female | 87 | 87 | 88 | 88 | 89 | 1.01 (0.99–1.02) | 0.482 |
| **Migrants** | 83 | 84 | 85 | 85 | 85 | 1.01 (0.99–1.02) | 0.142 |
| Male | 83 | 84 | 84 | 84 | 85 | 1.01 (0.99–1.01) | 0.286 |
| Female | 85 | 85 | 86 | 86 | 87 | 1.01 (0.98–1.03) | 0.569 |
| **Nonmigrant** | 93 | 93 | 94 | 94 | 94 | 1.00 (0.99–1.01) | 0.710 |
| Male | 93 | 93 | 93 | 93 | 93 | 1.00 (0.99–1.01) | 0.878 |
| Female | 94 | 93 | 93 | 93 | 92 | 0.99 (0.97–1.02) | 0.798 |
| **Male-to-male HIV exposure** | 90 | 90 | 91 | 91 | 91 | 1.00 (0.99–1.01) | 0.393 |
| Migrant | 83 | 84 | 84 | 84 | 84 | 1.00 (0.99–1.01) | 0.691 |
| Nonmigrant | 95 | 95 | 95 | 95 | 96 | 1.00 (0.99–1.01) | 0.483 |
| **Born in SEA** | 70 | 70 | 71 | 72 | 72 | 1.01 (0.99–1.03) | 0.369 |
| **Born in SSA** | 83 | 85 | 87 | 88 | 89 | 1.02 (0.99–1.04) | 0.137 |
| **RHCA eligible** | 94 | 95 | 95 | 96 | 96 | 1.01 (0.99–1.02) | 0.534 |
| **RHCA not eligible** | 80 | 81 | 81 | 82 | 83 | 1.01 (0.99–1.02) | 0.099 |
| **On ART (second 90)** | 72 | 74 | 77 | 86 | 90 | 1.06 (1.05–1.06) | <0.001 |
| Male | 70 | 73 | 76 | 86 | 91 | 1.06 (1.05–1.07) | <0.001 |
| Female | 66 | 67 | 71 | 76 | 78 | 1.04 (1.01–1.08) | 0.012 |
| Missing sex | 89 | 90 | 93 | 95 | 96 | 1.02 (0.99–1.04) | 0.122 |
| **Missing country of birth** | 70 | 73 | 75 | 91 | 94 | 1.08 (1.07–1.09) | <0.001 |
| **Migrant** | 71 | 72 | 78 | 81 | 85 | 1.04 (1.03–1.06) | <0.001 |
| Male | 67 | 69 | 75 | 79 | 84 | 1.05 (1.03–1.07) | <0.001 |
| Female | 69 | 69 | 75 | 77 | 79 | 1.03 (0.99–1.08) | 0.154 |
| **Nonmigrant** | 74 | 76 | 79 | 84 | 90 | 1.04 (1.03–1.05) | <0.001 |
| Male | 72 | 74 | 77 | 83 | 90 | 1.04 (1.03–1.05) | <0.001 |
| Female | 67 | 68 | 71 | 71 | 70 | 1.01 (0.95–1.07) | 0.707 |
| **Male-to-male HIV exposure** | 69 | 72 | 76 | 87 | 90 | 1.07 (1.06–1.08) | <0.001 |
| Migrant | 67 | 70 | 77 | 81 | 87 | 1.06 (1.04–1.08) | <0.001 |
| Nonmigrant | 73 | 75 | 79 | 85 | 92 | 1.05 (1.04–1.06) | <0.001 |
| **Born in SEA** | 76 | 77 | 83 | 83 | 87 | 1.03 (0.99–1.07) | 0.055 |
| **Born in SSA** | 80 | 83 | 88 | 90 | 93 | 1.03 (0.99–1.08) | 0.108 |
| **RHCA eligible** | 70 | 73 | 77 | 80 | 86 | 1.04 (1.01–1.07) | 0.016 |
| **RHCA not eligible** | 71 | 72 | 78 | 81 | 85 | 1.04 (1.02–1.06) | <0.001 |
| **Viral suppression (third 90)** | 89 | 91 | 92 | 95 | 95 | 1.02 (1.01–1.03) | <0.001 |
| Male | 89 | 91 | 92 | 95 | 95 | 1.02 (1.01–1.03) | <0.001 |
| Female | 84 | 87 | 87 | 91 | 91 | 1.02 (0.97–1.06) | 0.351 |
| Missing sex | 92 | 92 | 95 | 95 | 97 | 1.01 (0.99–1.04) | 0.331 |
| **Missing country of birth** | 91 | 94 | 95 | 96 | 97 | 1.01 (0.99–1.03) | 0.068 |
| **Migrants** | 87 | 89 | 90 | 93 | 93 | 1.02 (1.00–1.04) | 0.046 |
| Male | 85 | 88 | 88 | 93 | 92 | 1.02 (1.00–1.04) | 0.044 |
| Female | 85 | 88 | 86 | 91 | 90 | 1.02 (0.96–1.08) | 0.584 |
| **Nonmigrant** | 87 | 89 | 92 | 94 | 96 | 1.02 (1.01–1.04) | 0.001 |
| Male | 87 | 89 | 92 | 94 | 96 | 1.03 (1.01–1.04) | 0.001 |
| Female | 83 | 85 | 86 | 92 | 88 | 1.02 (0.95–1.09) | 0.609 |
| **Male-to-male HIV exposure** | 89 | 92 | 92 | 95 | 96 | 1.02 (1.01–1.03) | <0.001 |
| Migrant | 87 | 88 | 88 | 93 | 93 | 1.02 (0.99–1.04) | 0.076 |
| Nonmigrant | 87 | 90 | 92 | 95 | 96 | 1.02 (1.01–1.04) | 0.002 |
| **Migrants born in SEA** | 83 | 90 | 87 | 92 | 93 | 1.03 (0.99–1.06) | 0.178 |
| **Migrants born in SSA** | 88 | 91 | 92 | 92 | 91 | 1.01 (0.96–1.06) | 0.794 |
| **Migrants eligible for RHCA** | 90 | 91 | 91 | 96 | 95 | 1.02 (0.98–1.05) | 0.369 |
| **Migrants not eligible for RHCA** | 85 | 88 | 89 | 93 | 92 | 1.02 (0.99–1.04) | 0.064 |

Abbreviations: ART, antiretroviral therapy; IRR, incidence rate ratio; RHCA, reciprocal healthcare agreement; SEA, Southeast Asia; SSA, sub-Saharan Africa

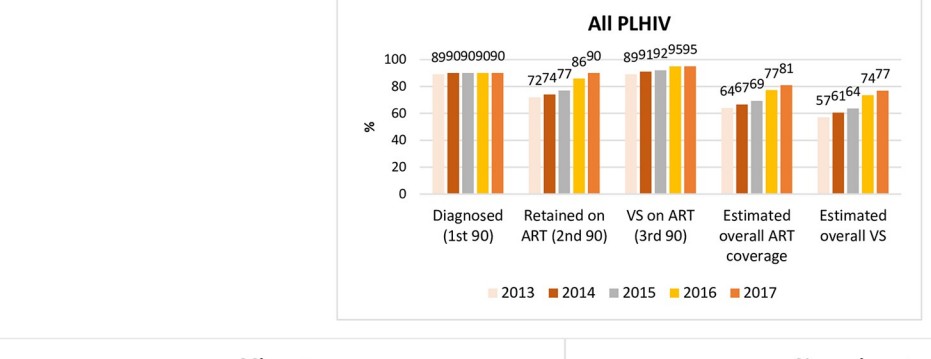

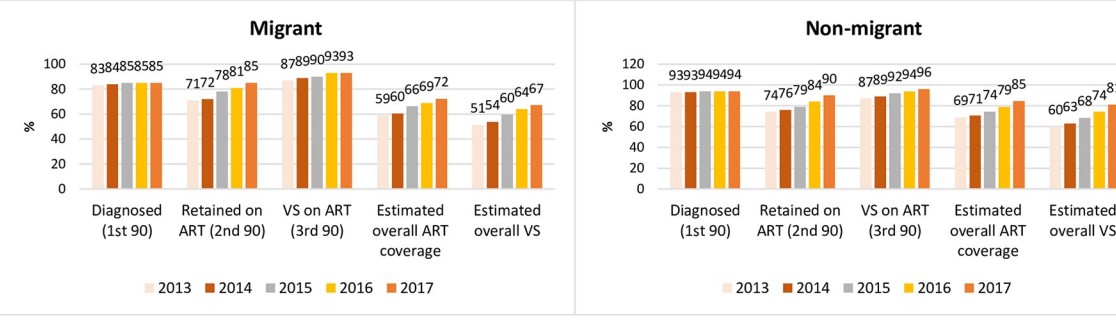

**Fig 1. HIV diagnosis and care cascades for all PLHIV, migrants, and nonmigrants in NSW and VIC (2013–2017).** ART, antiretroviral therapy; NSW, New South Wales; PLHIV, people living with HIV; VIC, Victoria; VS, viral suppression.

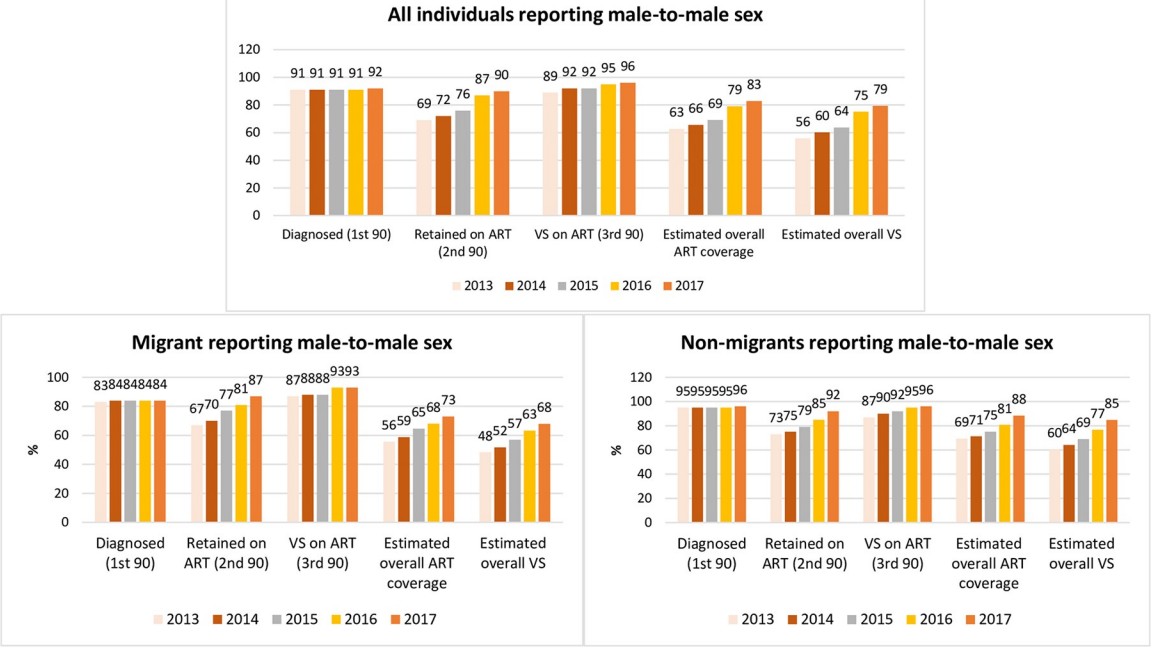

**Fig 2. HIV diagnosis and care cascades for all male-to-male HIV exposure, migrants, and nonmigrants in NSW and VIC (2013–2017).** ART, antiretroviral therapy; NSW, New South Wales; VIC, Victoria; VS, viral suppression.

## Discussion

Our results suggest that although gaps in the overall HIV diagnosis and care cascade in NSW and VIC decreased between 2013 and 2017, there were notable and sustained gaps in migrant subpopulations. Overall, migrants had lower HIV diagnosis and retention across the HIV care cascade than nonmigrants, especially the proportion diagnosed and on ART (first and second 90). Migrants reporting male-to-male HIV exposure, born in SEA and SSA, ineligible for subsidised healthcare through RHCA, and who were female had lower HIV diagnosis and retention across the care cascade than nonmigrants. There was a particularly low proportion of PLHIV born in SEA who were diagnosed (72% in 2017). As we approach 2020, the first Fast-Track target year, we estimated that in 2017, 70% and 65% of migrants estimated to be living with HIV in NSW and VIC were on ART and virally suppressed, respectively. This was lower than 80% and 77%, respectively, in nonmigrants estimated to be living with HIV.

To the best of our knowledge, this is the first analysis that examined and compared HIV diagnosis and the care cascade in migrants in Australia. In a survey of 55 European and Central Asian countries, Brown and colleagues (2018) identified that only 41% produced cascades for key and vulnerable populations, but there was wide variation between countries and no subpopulation comparisons [26]. Our findings are consistent with other studies that identified that mobile populations often face barriers to HIV testing and treatment access and often have lower VS, suggesting that they have lower HIV diagnosis and retention in the care cascade than citizens [26,27]. Italian researchers reported that migrants living with HIV had barriers to initiating ART and a higher risk of virologic failure compared with nonmigrants [28]. In countries with concentrated epidemics, HIV diagnosis and care among MSM are often similar to overall national cascades, but migrants often have lower treatment coverage and VS [26]. A review by Tanser and colleagues (2015) also identified poorer HIV diagnosis and care cascades among migrants and found that existing health system structures failed to properly account for migrants who still present late and have poor retention [29]. In Australia, the incidence of HIV in newly arrived Asian-born MSM was found to be higher than other subpopulations [30]. Although Medicare ineligibility is a significant barrier, it is known that migrants are less likely to be offered or less likely to initiate early ART despite having access [31,32].

Migrants are often more likely to have higher HIV prevalence, higher frequency of late HIV diagnosis, and lower VS rates and to experience barriers to healthcare access than the general population [29,33,34]. Studies in Australia have previously identified that migrants faced barriers to HIV testing, including cost and eligibility for health services, low visibility or awareness of HIV in Australia, HIV-related stigma, sociocultural and religious influences, and missed opportunities for testing during general practitioner visits [10,35]. Our study found that migrants from countries that are not eligible for RHCA had lower HIV diagnosis and care cascades compared with those who are eligible. An additional layer of barriers includes legal status of migrants. A study that compared documented and undocumented migrants in Italy found that despite free access to ART, migrants still had difficulties in gaining optimal HIV care [34]. In contrast, other studies found that when healthcare cost barriers are removed for migrants living with HIV, VS increases, thereby reducing risk of transmission [36,37]. Removing barriers that migrants living with or at risk of HIV face is expected to improve health outcomes and progress toward the UNAIDS Fast-Track targets.

The poorer cascade for Australian-born females is of particular concern. However, our analysis did not explore differences between indigenous and nonindigenous Australians. Surveillance data have shown that between 2013 and 2016, the rate of HIV notifications increased by 41% in the indigenous population and remained 1.6 times higher compared with the nonindigenous population in 2016 [6]. According to the surveillance data, a greater proportion of HIV notifications in

the indigenous population is attributed to heterosexual sex (21%) or injecting drug use (18%) compared with the Australian-born nonindigenous population (18% and 3%, respectively). Further comparisons between indigenous and nonindigenous Australian-born populations is warranted to compare whether there are differences in HIV testing, treatment coverage, and VS.

Previous work by our group suggests that reaching the Fast-Track targets may not be enough to achieve HIV incidence reduction, especially in countries like Australia with low HIV prevalence [4]. Scaling up other HIV prevention strategies, such as HIV testing, preexposure and postexposure prophylaxis, and condom use, would be required. Realising the full prevention and treatment benefits of the UNAIDS Fast-Track strategy will require reaching all key and vulnerable subpopulations, including migrants [29]. This is particularly important at this stage of the HIV epidemic response and with increasing levels of global migration [38]. Understanding the dynamics of HIV and migration can be complex because migrants can be at risk of HIV infection before, during, and after migration. HIV prevention and treatment programmes will need to take into consideration these dynamics with an inclusive approach.

Limitations of our study included a high proportion (38% in 2017) of individuals retained in ACCESS database missing a country of birth entry. This impacted our ability to categorise migrants or nonmigrants and migrant populations. However, we tracked ART coverage and VS among those missing country of birth and found similar or higher proportions compared with nonmigrants. We used a cross-sectional instead of a longitudinal approach to develop the cascades of care because it was not feasible to link individuals diagnosed in the National HIV Registry to the clinical database. Hence, we used the proportion of individuals retained in care in the clinical database who initiated ART to estimate the second 90 of the cascade. Another limitation was that all patients on ART, irrespective of ART duration, were counted in the analysis, including patients newly initiated on ART. This has the potential to overestimate the number not virally suppressed. The generalizability of our work is reduced because it only included two Australian states (NSW and VIC). However, these states have the highest proportion of PLHIV and migrants [7] and are likely to be reasonably representative of the national picture. Finally, the analysis considered country of birth as a proxy for migrant status and Medicare eligibility and may have misclassified people born in other countries who were Australian citizens and Medicare eligible. Additional data that can better classify migrants, including year of arrival, years lived in Australia, and visa category, are needed. Follow-up modelling studies are required to reveal the differences between migrants and nonmigrants in order to highlight additional sociodemographic characteristics that may be used to design public health programmes.

## Conclusion

Gaps in the overall HIV diagnosis and retention in the care cascade in NSW and VIC decreased between 2013 and 2017. However, there were notable and sustained gaps in migrant subpopulations. In particular, retention of migrants in the care cascade was lower compared with nonmigrants. Notable low retention occurred in migrants reporting male-to-male HIV exposure, migrants from SEA and SSA, migrants not eligible for subsidised healthcare access in Australia, and females (migrants and nonmigrants). Migration is a global issue, with global migration increasing. If the HIV response is to meet the Fast-Track HIV incidence reduction target, it is important to monitor HIV diagnosis and the care cascades in all subpopulations to ensure that no one is being left behind.

## Supporting information

**S1 STROBE Checklist.**
(DOC)

**S1 Text. Concept note.**
(DOCX)

**S1 Table. Estimated PLHIV diagnosed.** PLHIV, people living with HIV.
(DOCX)

**S2 Table. Retained PLHIV on ART.** ART, antiretroviral therapy; PLHIV, people living with HIV.
(DOCX)

## Acknowledgments

We would like to acknowledge people living with HIV who contributed to this research and the clinicians and surveillance officers in the different health jurisdictions in Australia. We are grateful to the Australian HIV Cascade Reference Group and the ACCESS team, who provided their input to this research.

## Author Contributions

**Conceptualization:** Tafireyi Marukutira, Richard T. Gray, Suzanne Crowe, Rebecca Guy, Mark Stoove, Margaret Hellard.

**Data curation:** Tafireyi Marukutira, Caitlin Douglass, Clarissa Moreira, Jason Asselin, Basil Donovan, Tobias Vickers.

**Formal analysis:** Tafireyi Marukutira, Richard T. Gray, Caitlin Douglass, Jason Asselin, Tobias Vickers, Tim Spelman.

**Investigation:** Tafireyi Marukutira.

**Methodology:** Tafireyi Marukutira, Richard T. Gray.

**Supervision:** Richard T. Gray, Tobias Vickers, Tim Spelman, Suzanne Crowe, Rebecca Guy, Mark Stoove, Margaret Hellard.

**Validation:** Tafireyi Marukutira, Richard T. Gray, Carol El-Hayek, Jason Asselin, Basil Donovan, Tobias Vickers.

**Visualization:** Tafireyi Marukutira, Caitlin Douglass.

**Writing – original draft:** Tafireyi Marukutira, Caitlin Douglass.

**Writing – review & editing:** Tafireyi Marukutira, Richard T. Gray, Caitlin Douglass, Carol El-Hayek, Clarissa Moreira, Jason Asselin, Basil Donovan, Tobias Vickers, Tim Spelman, Suzanne Crowe, Rebecca Guy, Mark Stoove, Margaret Hellard.

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
