## [Decision Letter · Decision Letter 0]

15 Dec 2019

Dear Dr. Marukutira,

Thank you very much for submitting your manuscript "HIV diagnosis and care cascade for migrants in Australia : Identifying gaps and opportunities towards the last mile to HIV elimination" (PMEDICINE-D-19-03230) for consideration at PLOS Medicine for our upcoming special issue on refugee and migrant health. 

Your paper was evaluated by the editors and sent to independent reviewers, including a statistical reviewer. The reviews are appended at the bottom of this email and any accompanying reviewer attachments can be seen via the link below:

[LINK]

In light of these reviews, we will not be able to accept the manuscript for publication in the journal in its current form, but we would like to invite you to submit a revised version that fully addresses the reviewers' and editors' comments. You will appreciate that we cannot make a decision about publication until we have seen the revised manuscript and your response, and we expect to seek re-review by one or more of the reviewers. 

We hope to receive your revised manuscript by January 3rd. Please email us (plosmedicine@plos.org) if you have any questions or concerns.

Please let me know if you have any questions. Otherwise, we look forward to receiving your revised manuscript soon. 

Sincerely,

Richard Turner, PhD

rturner@plos.org

Noting referee 1's comments, PLOS Medicine policy is that significance tests should be performed and reported with 95% CI and p values where appropriate. 

Please adapt the title so that the segment after the colon consists predominantly of a study descriptor, e.g., "HIV Diagnosis and Care Cascade for Migrants in Australia: a cross-sectional study", and include study date(s). 

We ask you to combine the "methods" and "findings" subsections of your abstract. Please add a new final sentence to the new combined subsection to summarize the study's main limitations.

Please add summary demographic and country or region of origin details for study participants to abstract and main text. 

After the abstract, we will need to ask you to add a new and accessible "author summary" section in non-identical prose. You may find it helpful to consult one or two recently published research papers in PLOS Medicine to get a sense of the preferred style. 

Early in the methods section, please state whether or not the study had a protocol or prespecified analysis plan, and if so attach the document(s) as a supplementary file (referred to in the text). Please highlight analyses that were not prespecified. 

Please avoid claims of "the first", and where used (e.g. currently at the start of the Discussion section) please add "to our knowledge" or similar. 

The first paragraph of the discussion should summarize the study's findings, with elements of discussion being restricted to subsequent paragraphs. 

Please ensure that all references meet journal format. For example, "and" should be removed from the author list for reference 4, and journal names should be abbreviated as appropriate. 

Please correct "report" at line 410. 

Please add a completed checklist for the most appropriate reporting guideline, which might be STROBE or RECORD, as a supplementary document (referred to in the methods section). In the checklist, please refer to individual items by section (e.g., "Methods") and paragraph number rather than by page or line numbers, as the latter generally change in the event of publication. 

Comments from the reviewers:

*** Reviewer #1:

 This is an important study and shows important information and I think the authors have done a nice job with the study. There are a few places where I think the reporting is unclear and needs some revising and possibly the methods, but I think these changes are not overly major. Below are specific comments for the authors.

I do want to commend the authors on the reporting of precisely null results! This is important to know, but often gets confused with non-significance. I would agree your findings are null even though you are near significance! My only quibble would be to ask you not to use significance testing at all and remove reference to "with the significance level set at p<0·05" in the methods and remove the pvalues from the results.

In the methods, can you clarify this statement, "To estimate the number of people living with diagnosed HIV (PLDHIV) we used New South Wales (NSW) and Victorian (VIC) data from the Australian National HIV Registry, enhanced by surveillance among people recently diagnosed with HIV to improve migration estimates." What does it mean to be "enhanced"?

You note that "The ACCESS network collects demographic, testing, diagnosis and treatment data to monitor the sexual health of high‑ risk populations. Compared to other states, NSW and VIC have the highest coverage in the ACCESS database and they comprise the highest proportion of PLHIV and migrants in Australia [6, 17]." But it is important to state here how high the coverage is. Is this the 52% and 61% percent mentioned in the next few sentences? This wasn't clear to me.

I found the explanation of how the first two steps of the cascade were created to be unclear. I begins by talking about mean matching imputation but I am not clear on what is being imputed. Is this the number of people who are infected and if so, based on what data? Is it the registry data and if so, why does this need to be imputed? Or is the idea you are just imputing covariates within the registry data? It might help to start that paragraph within "In order to [state the objective]…" statement to make it clearer why the imputation was being done. And if this was based on notifications, wouldn't this be persons living with HIV who know their status and not all people with HIV?

I was confused about the second 90 which appears to be limited to those in care. This doesn't seem right as the second 90 is meant to be everyone on ART and the third 90 should be everyone who is virally suppressed. Limiting to just those in care would seem to inflate the numbers and also not carry through everyone in the original denominator of 100%. I could be misunderstanding this but that is the way it read to me. 

The reason I think I could be wrong is the next section says "Proportion of PLDHIV retained in care on ART (2nd 90)" So if I'm wrong about the above, it isn't clear why this is the 2nd 90 as that would usually be the proportion on ART. Perhaps these are mathematically equivalent in which case I would just make that clearer.

The validity of the second and third 90 really depends on how representative the ACCESS database is. Can you provide any information on that given it appears to only be 50% complete?

*** Reviewer #2: 

I confine my remarks to statistical aspects of this paper. The basic approach is fine, but I have some corrections and suggestions before I can recommend publication.

First, I wonder why the authors didn't model the differences between groups that they seem interested in. It isn't *necessary* to do so - and the title of the paper is not about modeling - but they do seem interesting. Maybe that would be a different paper.

Line 24 Perhaps I am confused, but this seems to contradict the result reported in the results section.

Line 25-28 and similarly in the results section (line 249-254) - all of these rates did increase, just not significantly. Part of this can be corrected by adding the word "significantly" but the authors should be careful not to accept the null. And, if they want to comment on statistics about differences between groups, they should model them (e.g. with a regression). 

Line 196-197 I don't understand how these plausible ranges were found or what they were used for. 

Table 2 - I would get rid of the CIs in the cells for years. It's not that they are wrong, but they aren't needed and they make the table very hard to read.

Fig 1 and 2: I would use line graphs with time on the x axis and % on the y axis. I would have 5 graphs in each figure - one for each "90" - each with3 lines (one for all, one for migrants and one for non-migrants) (or maybe no line for "all")

Peter Flom

*** Reviewer #3: 

A well written paper describing the differential treatment cascades within the Australian system, according to migrant status. Although it is not essential the authors might consider a few additions to help contextualize the paper, especially for overseas readers; Firstly, it should be noted that 28.5 % of the Australian general population is overseas born (according to the 2016 census) so the proportion of PLHIV is consistent with those figures. Secondly, the CD4 back projection method, to estimate the number of undiagnosed PLHIV may not be as accurate when looking at sub populations for countries over represented in late presentations (mainly due to cultural or religious stigma). 

Additionally, although the focus of the paper is on the apparent inequality of health outcomes for migrants, there is no separation of indigenous vs non indigenous status in the Australian born group. The poorer cascade for Australian born females suggests this may be an important issue.

Finally, the assumption that overseas born PLHIV is a surrogate for inability to access care is a bit of an over simplification. Many overseas born people do have PR or residency and can access Medicare as well as anyone else. For those without Medicare, all sexual health clinics provide free care and can access free antiviral medication now. It would seem that the key barrier is reluctance of the migrant to access testing or care, rather than these services are not available to them. this is reflected in the increasing rates of on therapy and VS in all groups but not in any improvement in the proportion diagnosed.

***

[LINK]

---

## [Decision Letter · Decision Letter 1]

14 Jan 2020

Dear Dr. Marukutira,

Thank you very much for re-submitting your manuscript "Gaps in the HIV diagnosis and care cascade for migrants in Australia: A cross-sectional study (2013-2017)" (PMEDICINE-D-19-03230R1) for consideration at PLOS Medicine for our upcoming special issue on refugee and migrant health.

I have discussed the paper with editorial colleagues and the guest editors for the special issue, and it was also seen again by one reviewer. I am pleased to tell you that, provided the remaining editorial and production issues are dealt with, we expect to be able to accept the paper for publication in the journal.

[LINK]

We hope to receive your revised manuscript within around one week. Please email us (plosmedicine@plos.org) if you have any questions or concerns.

Please let me know if you have any questions. Otherwise, we look forward to receiving the revised manuscript soon. 

Kind regards,

Richard Turner, PhD

rturner@plos.org

Requests from Editors:

Please adapt the title slightly: we suggest "Gaps in the HIV diagnosis and care cascade for migrants in Australia, 2013-2017: a cross-sectional study".

Please revise your abstract, aiming to improve readability of the "methods and findings" subsection. 

At line 4 in your abstract, we suggest adding a few words to quote the UNAIDS targets. 

At line 24 (and line 246 in the main text), please quote the number or proportion of males, say, in the total cohort. 

To the final sentence of the "methods and findings" subsection of your abstract, please add 1-2 further limitations. We ask you to include the point made at line 379, i.e., that estimation of the "second 90" was done by assessing those in care. 

Please restructure the "author summary" so that it consists of the existing three sections, but with each section comprising 3-4 bulleted points. The individual points should consist of one sentence, in general. Please consult one or two recent research papers published in PLOS Medicine to get a sense of the preferred style. 

At line 254 and elsewhere in the ms, please substitute "(Table 1)". In general, square brackets should only be used inside parentheses.

At line 291, please remove "by 6%" (which is implicit in the point estimate) to leave "... increased (IRR 1.06 ... )".

Please add initial zeroes to the p values in table 2. 

At line 358, should that be "surveillance data"?

Throughout the text, please adapt the formatting of reference callouts as follows: "... risk of transmission [36,37].".

In the reference list, please ensure that journal names are abbreviated as appropriate (e.g., reference 4). 

Please remove ampersands from the reference list (e.g., reference 10). 

Please correct the spelling of the journal name for reference 15. 

To the attached STROBE checklist, please add paragraph numbers, where applicable, to the rightmost column. 

Comments from Reviewers:

*** Reviewer #2: 

The authors have addressed my concerns and I now recommend publication

Peter Flom

***

[LINK]

---

## [Editor Report · Decision Letter 2]

31 Jan 2020

Dear Dr. Marukutira, 

On behalf of my colleagues and the academic editor, Dr. Paul Spiegel, I am delighted to inform you that your manuscript entitled "Gaps in the HIV diagnosis and care cascade for migrants in Australia, 2013-2017: A cross-sectional study" (PMEDICINE-D-19-03230R2) has been accepted for publication in PLOS Medicine. 

PRODUCTION PROCESS

PRESS

PROFILE INFORMATION

Thank you again for submitting the manuscript to PLOS Medicine. We look forward to publishing it. 

Best wishes, 

Richard Turner, PhD

Senior Editor 

PLOS Medicine

plosmedicine.org